Sentiment analysis of vegan related tweets using mutual information for feature selection

Shamoi Elvina 1
Turdybay Akniyet 1
Shamoi Pakizar p.shamoi@kbtu.kz 1
Akhmetov Iskander iskander.akhmetov@gmail.com 1 2
Jaxylykova Assel 2 3
Pak Alexandr 1 2
1 School of Information Technology and Engineering, Kazakh-British Technical University , Almaty , Kazakhstan
2 Institute of Information and Computational Technologies , Almaty , Kazakhstan
3 Kazakh National University , Almaty , Kazakhstan
Reyes-Menendez Ana
Electronic publication date: 2022 Dec 5
Publication date: 2022
Volume: 8
Electronic Location ID: e1149
Received 2022 Jun 10; Accepted 2022 Oct 17
Copyright: ©2022 Shamoi et al.
Copyright year: 2022
Copyright holder: Shamoi et al.
License: This is an open access article distributed under the terms of the Creative Commons Attribution License, which permits unrestricted use, distribution, reproduction and adaptation in any medium and for any purpose provided that it is properly attributed. For attribution, the original author(s), title, publication source (PeerJ Computer Science) and either DOI or URL of the article must be cited.
License URL: https://creativecommons.org/licenses/by/4.0/

Keywords: Natural language processing, Sentiment analysis, Emotion detection, Social media monitoring, Opinion mining, Twitter sentiment classification, Vegan tweets, Pointwise mutual information, Sentiment tracking

Funding: The Ministry of Education and Sciences of the Republic of Kazakhstan #AP09260670 #AP14871214 This work was supported by the Ministry of Education and Sciences of the Republic of Kazakhstan under the following grants: #AP09260670 “Development of methods and algorithms for augmenting input data for modifying vector word embeddings” and #AP14871214 “Development of machine learning methods to increase the coherence of text in summaries produced by the Extractive Summarization Methods”. The funders had no role in study design, data collection and analysis, decision to publish, or preparation of the manuscript.

==============================
Nowadays, people get increasingly attached to social media to connect with other people, to study, and to work. The presented article uses Twitter posts to better understand public opinion regarding the vegan (plant-based) diet that has traditionally been portrayed negatively on social media. However, in recent years, studies on health benefits, COVID-19, and global warming have increased the awareness of plant-based diets. The study employs a dataset derived from a collection of vegan-related tweets and uses a sentiment analysis technique for identifying the emotions represented in them. The purpose of sentiment analysis is to determine whether a piece of text (tweet in our case) conveys a negative or positive viewpoint. We use the mutual information approach to perform feature selection in this study. We chose this method because it is suitable for mining the complicated features from vegan tweets and extracting users’ feelings and emotions. The results revealed that the vegan diet is becoming more popular and is currently framed more positively than in previous years. However, the emotions of fear were mostly strong throughout the period, which is in sharp contrast to other types of emotions. Our findings place new information in the public domain, which has significant implications. The article provides evidence that the vegan trend is growing and new insights into the key emotions associated with this growth from 2010 to 2022. By gaining a deeper understanding of the public perception of veganism, medical experts can create appropriate health programs and encourage more people to stick to a healthy vegan diet. These results can be used to devise appropriate government action plans to promote healthy veganism and reduce the associated emotion of fear.

Introduction

A vegan diet (or plant-based diet) assumes refraining from consuming animal or animal-derived products, such as meat, fish, dairy, eggs, cheese, etc., (MacInnis & Hodson, 2017). Fruits, vegetables, beans, whole grains, legumes, nuts, and a range of other whole plant foods make up this diet. The motivations for switching to a plant-based diet usually fall into one of the three categories: health benefits, animal rights, and environmental protection.

Vegans have traditionally been portrayed negatively in society and social media. There is even a term to describe this opposition: vegaphobia (Cole & Morgan, 2011). Vegaphobic sentiments and bias towards vegans appeared in the 2010s (MacInnis & Hodson, 2017).

However, in recent years, the COVID-19 pandemic and global warming have increased the awareness of plant-based lifestyles and indeed produced significant changes in people’s daily lives (Park & Kim, 2022; WHO, 2021; Rzymski et al., 2021; Alae-Carew et al., 2022). Moreover, in 2015, based on numerous studies, the International Agency for Research on Cancer (IARC) of the World Health Organization (WHO) declared that processed meat is a group 1 carcinogen and that red meat is a group 2 carcinogen (Willett et al., 2019). At the same time, several recent studies (Dagevos & Voordouw, 2013; Willett et al., 2019) report that a plant-based diet is better for the environment and more sustainable than other kinds of diet. Because of all those reasons, combined with benefits for human health and environmental sustainability, interest in a plant-based diet is expanding. These days of climate change, this subject is becoming increasingly significant and is frequently debated on social media.

Besides sustainability issues, one of the key motives for following a vegan diet is for health reasons (Radnitz, Beezhold & DiMatteo, 2015; Jennings et al., 2019). A vegan diet has been linked in several studies with better health and longevity, reduced blood pressure and cholesterol, lower rates of heart disease, type 2 diabetes, and various forms of cancer (Willett et al., 2019; Norman & Klaus, 2019; Qian et al., 2019). Globally, there is a rise in the number of people with obesity, which is associated with many serious medical problems (Finucane et al., 2011; Revels, Kumar & Ben-Assuli, 2017). Moreover, research indicates that obesity is linked to a higher risk of severe COVID-19, cardiovascular disease, and cancer (Finucane et al., 2011). At the same time, studies prove that a plant-based diet helps individuals to lose weight and prevent obesity (Tran et al., 2020; Turner-McGrievy, Mandes & Crimarco, 2017).

Several studies aimed to find the association between diet and eating habits with the risk and severity of COVID-19 (Aman & Masood, 2020; James et al., 2021; Zabetakis, Matthys & Tsoupras, 2021; Kim et al., 2021; Merino et al., 2021). Plant-based diets were linked to a lower risk of moderate-to-severe COVID-19 (Kim et al., 2021). Another study revealed that a diet rich in plant-based foods had been linked to reduced risk and severity of COVID-19 (Merino et al., 2021). Therefore, a vegan diet may offer protection against severe COVID-19.

Another study (Di Renzo et al., 2020) reports that during the COVID-19 pandemic, population lifestyle and eating habits changed. Specifically, weight gain was observed in 48.6% of the respondents; 3.3% decided to stop smoking; 38.3% reported an increase in physical activity; 15% of the participants started purchasing organic fruits and vegetables, and many people aged 18-30 years shifted to the Mediterranean diet. These findings are also supported by another study (Singh, Jain & Rastogi, 2021), which reported a substantial decrease in consumption of junk food (73.8%), alcohol (27.6%), and smoking (8.1%).

Previous research has revealed that the number of people following a vegan or vegetarian diet is increasing (López et al., 2019). Another study showed that individuals regard vegan, organic, and homemade foods as the healthiest options (Pila, Kvasnikov Stanislavsk & Kvasnika, 2021). The three basic characteristics of food that were most communicated in 2020 were vegan, homemade, and organic.

Many people’s daily lives are now dominated by social media (Ostic et al., 2021). People use social networks to connect with other people, share opinions on various subjects, study and work. Understanding people’s views, experiences, and attitudes through social media analysis can play a vital role in government action program development (Pila, Kvasnikov Stanislavsk & Kvasnika, 2021). In turn, this is crucially important in promoting healthy eating, tackling global warming, and lowering a person’s COVID-19 risk, among others. One of the social media to express people’s opinions is Twitter. Twitter is one of the most widely used social media (Williams, Terras & Warwick, 2013) that allows users to write everything that comes to their mind in the form of tweets (Williams, Terras & Warwick, 2013).

The present article aims to utilize Twitter posts to understand public opinion on social media better as expressed on Twitter regarding the vegan (plant-based) diet from 2010 to 2022. We also plan to identify what was the impact of the pandemic on adherence to a vegan diet specifically and the main topics (words) when veganism is discussed in a positive or negative context. The study employs a data set containing a collection of popular vegan-related tweets.

As we see, previous studies on plant-based diets were primarily survey-based and conducted from April to May 2020. On the other hand, we use Twitter data, not surveys, to examine how public perception of veganism changed over a 12-year period, whether people felt positive or negative about vegan diets, and the prevailing emotions behind those sentiments.

We seek to answer the following research questions using vegan tweets from 2010–2012:

• How has the public perception of veganism changed over the past 12 years?

• What were the prevailing emotions associated with this change?

• Did the global pandemic of COVID-19, climate crisis, vegan documentaries, and research unveiling the health benefits of the vegan diet affect the attitude toward the vegan diet on Twitter?

The article presents the first comprehensive analysis of the public perception of the vegan diet from 2010 to 2022 on Twitter with an emphasis on new insights into the key emotions associated with this growing trend. Association with the key factors influencing this trend throughout the period was also done. Considering that:

• In 2018, UN IPCC reported that climate change catastrophe would occur in the upcoming 12 years (Masson-Delmotte et al., 2019);

• A plant-based diet is more sustainable and better for the environment than other diets, according to recent studies (Dagevos & Voordouw, 2013; Willett et al., 2019; Scarborough et al., 2014). For example, by comparing beef and soybeans, one study investigated that soy needs 18 times less energy per gram of protein and produces 71 times less CO2 than beef (Cleveland & Gee, 2017). Another study indicates that the vegan diet has the lowest CO2 emission compared to other diets (Chai et al., 2019). An Oxford University study reports that meat-eaters are responsible for about two and a half times as much dietary greenhouse gas emissions per day as vegans (Scarborough et al., 2014);

• Environmental sustainability in these days of climate change is of primary importance (Masson-Delmotte et al., 2019; Dagevos & Voordouw, 2013),

this topic is of vital significance. By gaining a better understanding of the opinion on veganism and associated emotions, medical professionals can design effective health initiatives and encourage more people to either stick to a healthy plant-based diet or reduce animal product consumption. Even small dairy and meat reductions in many people may make a valuable contribution to climate change (Scarborough et al., 2014). According to recent research on the quantitative estimate of the potential climate effects of a hypothetical transition to a plant-based diet worldwide, such a phaseout can stabilize greenhouse gas levels for 30 years and offset 68% of CO2 emissions this century Eisen & Brown (2022) (see Fig. 1).

We aim to use sentiment analysis to track the change in public perception of veganism. Sentiment analysis is a computational study for identifying emotions represented in texts. The purpose of sentiment analysis is to classify the texts’ polarity—determine whether a piece of text (tweet in our case) conveys a negative or positive viewpoint (Neethu & Rajasree, 2013; Chinnalagu & Durairaj, 2021). Therefore, sentiment analysis is mainly used to grasp opinions about trends, products, services, or political issues.

Although there are various approaches to constructing sentiment analysis models, we chose the point-wise mutual information (PMI) approach proposed by Church & Hanks (2002), Turney & Littman (2003a), Turney & Littman (2003b) and Turney (2002) for its interpretability and robustness to statistical bias in small sample sizes. Another motivation for using the mentioned approach is its application to perform feature selection in this study. MIT researcher Robert Fano introduced the basics of the approach concept in 1961 (Fano, 1961) and Jurafsky et al. (2014) initially referred to as mutual information (MI). The PMI or MI in NLP application measures the probability of two-word co-occurrence relative to the random probability, thus giving more sense to the semantic proximity of the words. Therefore, our rationale behind using the method is to see how effectively the words sought to be associated with certain sentiment classes (PMI measures) can serve as features in training the sentiment classifier model. A number of studies use the PMI-based approach to sentiment analysis (Bindal & Chatterjee, 2016; Vo & Ock, 2012; Feldman, 2013; Hamdan, Bellot & Bechet, 2015; Utama, 2019; Bonzanini, 2016; Kanna & Pandiaraja, 2019).

Figure 1 Global phaseout of animal agriculture and transition to a plant-based diet can potentially reduce global warming’s impact on the atmosphere.

Reprinted from Eisen & Brown (2022).

The article is structured as follows. Section I is this introduction. An overview of the sentiment analysis techniques is presented in Section II. In Section III, we describe our method—the Pointwise Mutual Information-Based Feature Selection for Sentiment Analysis. In addition, we perform emotion analysis at this stage to take a more in-depth look at the public’s interest in veganism. Data collection and data annotation procedures are also described there. Evaluation of our method along with experimental results is shown in Section IV. Finally, concluding remarks are provided in Section V. The last section also presents insights on how the method can be improved in the future.

The following section provides an overview of sentiment analysis techniques.

Sentiment Analysis: Theoretical Background

Approaches for sentiment analysis

The unprecedented growth of user-generated content that we are witnessing nowadays on the World Wide Web (WWW) implies that for the first time in humankind’s history, the broad public can express their opinions on various subjects almost effortlessly. Of course, this happened with the help of contemporary technologies and Social Network (SN) phenomena emerging in the early 2000s. And analyzing public opinions on subjects of interest is vital for government and business.

Sentiment analysis (SA) or opinion mining (OM) is identifying and classifying people’s attitudes toward a subject or entity using the quantitative methods of statistics, computational linguistics, and machine learning (ML) (Neethu & Rajasree, 2013). The sentiment analysis topic has been a subject of rigorous research in the last years (Ligthart, Catal & Tekinerdogan, 2021), and the beginning of the research in this area can be traced back to the middle of the 20th century (Sims, 2015).

SA tasks can be divided into the following major categories (Hemmatian & Sohrabi, 2019; Ligthart, Catal & Tekinerdogan, 2021):

1. Opinion extraction is the process of user opinion extraction from WWW to discover their way of thinking.

2. Sentiment mining is the separation of objective or factual sentences from the subjective ones that contain opinions and sentiments.

Subjectivity classification attributes a sentence to a class of subjectivity it carries.

3. Sentiment classification categorizes the opinions discovered into predefined classes.

Polarity determination whether it is positive, negative, or lacks the polarity (neutral).

The approaches for sentiment analysis can be classified as follows (Adwan et al., 2020); see also Fig. 2:

Figure 2 Sentiment analysis methods classification.

1. Lexicon based approaches use dictionaries (Hu & Liu, 2004) of words that are assumed to be positive, negative, or neutral, and then weigh or score the text according to the relative content of these words to infer its sentiment (Akilandeswari & Jothi, 2018; Das et al., 2018; Azizan et al., 2019).

Dictionary-based approaches use a vocabulary look-up technique to detect sentiment-rich words and their polarity.

Corpus-based approaches work with the statistical models developed on many texts to leverage the information on the context of word use and the dictionary-based approach.

2. ML approaches can be subdivided into Supervised and Unsupervised Learning classes of tasks:

Supervised learning methods require a dataset with items labeled for the sentiment to train classification modeling algorithms (SVM, KNN, DTC, LR, and others).

Unsupervised learning methods search for the structure of information within the presented data using clustering, topic modeling, and mapping algorithms (Mukhamediev et al., 2020; Kirill et al., 2020).

3. Hybrid approaches, as the name implies, use both the Lexicon-based and ML approaches together.

A challenging task in sentiment analysis is the domain specificity of the developed approaches, which come from the fact that sentiment-rich words can carry different polarity in usage across domains. For example, the term “aggressive” can be perceived as positive when related to sports (“Chelsea played aggressively tonight in a match with Barcelona club.”) and negative in the political domain (“Russian troops aggressively attack Mariupol city in Ukraine, thousands of civilians get injured.”). Thus, topic-aware or domain-specific models are advantageous over the general ones (Akhmetov, Gelbukh & Mussabayev, 2022).

Another problem that definitely should be addressed is the possible data imbalance on the sentiment classes, especially in the domains where most publications are negative (war, accident, crime news). Handling the data imbalance can be done with under-over-sampling techniques, data augmentation, and others (Omarkhan, Kissymova & Akhmetov, 2021; Kelsingazin, Akhmetov & Pak, 2021).

One of the classical works on the Twitter micro-blog post (tweet) analysis is VADER (Hutto & Gilbert, 2014), a simple rule-based sentiment analysis model. VADER employs an empirically validated gold-standard lexical feature list with the associated sentiment or opinion polarity and specializes in micro-blog short text messages. The model shows an F1 classification accuracy of 0.96, outperforming human raters with a 0.84 result. Another model trained on Twitter data (Kocaman & Talby, 2021) shows an accuracy of 0.80.

Related works

Analyzing posts in SN can be of particular interest to government administration, business officials from the marketing and sales side, social scientists, and anthropologists, as they contain vast amounts of information characterizing the post writers’ origins, cultural background, political views, and attitudes to a broad spectrum of objects, events and more abstract things as ideology.

One of the most popular SN is Twitter: the micro-blogging platform, where its 1.3 billion users make 500 million tweets every day (Alhgren, 2022). Analyzing Twitter posts or the so-called tweets involves exploring the interest, thoughts, and opinions in various aspects and domains (Adwan et al., 2020).

The application examples of tweet analysis are to understand people’s opinions towards different topics such as COVID-19, disasters, presidential elections, armed conflicts across the Globe (Ukraine, Syria, Palestine, Iraq), sports, and healthy lifestyles. Much research has focused on mining public opinion on various social, cultural, economic, and environmental topics on Twitter (Reyes-Menendez, Saura & Alvarez-Alonso, 2018; Reyes-Menendez, Saura & Filipe, 2020; Saura, Reyes-Menendez & Palos-Sanchez, 2019; Serrano et al., 2021; Das et al., 2018; Pila, Kvasnikov Stanislavsk & Kvasnika, 2021; Islam, 2019; Gore, 2015; Karami et al., 2018; Pila, Kvasnikov Stanislavsk & Kvasnika, 2021; Saura, Palos-Sanchez & Rios Martin, 2018). Reyes-Menendez, Saura & Alvarez-Alonso (2018) investigated the main factors that worry people about the environment and public health. Analysis of health-related tweets by Islam (2019) revealed an interesting correlation between yoga and veganism. Gore (2015) studied geographic issues of obesity rate in America. The main characteristics of public perceptions of obesity, diabetes, nutrition, and exercise were explored by Karami et al. (2018). Reyes-Menendez, Saura & Filipe (2020) identified the main challenges involved in the MeToo movement. In Saura, Reyes-Menendez & Palos-Sanchez (2019) authors explored negative and positive perceptions of consumers related to the Black Friday event. Next, Serrano et al. (2021) examines online reviews of green Airbnb users to find the main aspects among which the term sustainability appears. Pila, Kvasnikov Stanislavsk & Kvasnika (2021) explored all tweets between 2019 and 2020 that included the hashtag “healthyfood”.

Methods

Data collection

We study Twitter messages or tweets related to veganism to gain insight into the change in the sentiment of such posts during the COVID-19 world pandemic, before it, and just after it, including publication of studies and documentaries about veganism.

Kaggle dataset

For Sentiment analysis classification model training, we used the Sentiment140 dataset (https://www.kaggle.com/datasets/kazanova/sentiment140) with 1.6 million year 2009 tweets labeled for sentiment (0 = negative, and 4 = positive). We filtered the dataset by selecting only tweets containing the following keywords: “vegan”, “plantbased”, “healthy”, “vegetarian”, “veggie”, “veganism”, “cruelty free”, “plant milk”, “beyond meat”.

After filtering the tweets by vegan keywords, we ended up with a dataset of 1,876 tweets containing our target keywords. The initial dataset contained equal amounts of positive and negative labels or 800K each. However, the filtered tweets include 798 negative and 1,078 positive tweets. Thus, initially, the vegan tweets are more positive: the mean sentiment is 2.3 as opposed to the general score of 2.0 for all the tweets in the original dataset.

The most popular vegan keyword is “healthy” occurring in 955 of the selected tweets, followed by “veggie” (391), “vegan” (364) and “vegetarian” (191). For some of the keywords (“plantbased”, “plant milk” and “beyond meat”), we found no tweets, and 31 tweets possess more than one keyword; see Table 1.

Scraping the twitter

For this study, data was collected from Twitter between January 1st, 2010, and May 19th, 2022, using Python snscrape module (JustAnotherArchivist, 2022). The search was performed using the following keywords: “vegan”, “vegetarian”, “veggie”, “veganism”, “plant-based”, “cruelty-free”, “plant milk”, “tofu”. We refer to a keyword as a word that is present either in the form of a hashtag or anywhere in the body of a tweet. As we have seen in the Kaggle dataset, out of 1.6 million tweets, the tweets containing at least one of the keywords amounted to 1,876, which means a 0.0011725 content ratio. Thus, assuming 500 million tweets daily, we would have approximately 586,250 of our target tweets. Therefore, for the sake of getting the data fast, we limited the collection of data by 100 tweets for every keyword monthly along the 136 months and 19 days period collecting 113,677 tweets in total.1

Among the collected tweets, 87% were in English, 3% in Japanese, and 2% in Spanish. We have also labeled tweets in other languages with less than 1000 tweets as “other”; additionally, Twitter has a category “und” for undefined languages; see Fig. 3. Besides English, there are numerous Japanese tweets, which signal that the vegan topic is trendy in Japan.

Table 1 Kaggle Sentiment140 dataset vegan keyword content.

Keyword	Tweet count	
vegan	364	
plantbased	0	
healthy	955	
vegetarian	191	
veggie	391	
veganism	5	
cruelty free	1	
plant milk	0	
beyond meat	0	

Figure 3 Scraped tweets share by language.

From Table 2, we see that 70% of the tweets do not have a like, reply, retweet, or quote and are entirely unnoticed. As for the most popular tweets, they comprise the top 10% by replies/retweets or having at least one reply or retweet, the top 30% by having at least one like, and the top 1% by having at least one quote.

Table 2 A number of replies, retweets, likes, and quotes in the scraped tweets.

	Replies	Retweets	Likes	Quotes	
count	113,677	
mean	0.33	0.53	3.14	0.05	
std	3.92	23.02	344.35	4.20	
min	0.00	0.00	0.00	0.00	
50%	0.00	0.00	0.00	0.00	
70%	0.00	0.00	1.00	0.00	
80%	0.00	0.00	1.00	0.00	
90%	1.00	1.00	3.00	0.00	
95%	1.00	2.00	5.00	0.00	
97%	2.00	3.00	9.00	0.00	
99%	3.00	7.00	28.00	1.00	
max	1,119.00	7,492.00	115,499.00	1,398.00	

Taking into account that the majority of the tweets have no likes, we have decided to analyze only the tweets that have at least one like. Thus, we have 36,959 tweets to explore; see Fig. 4 for the distribution of the liked tweets by the query.

After filtering out the tweets with no likes, we observed in the Fig. 5 that there had been a steady rise in the number of tweets on the “vegan” topic since 2012 from less than 1,000 tweets per year to more than 5,000 tweets in 2021. This can be explained by the fact, that starting from 2011, a large amount of research has begun to appear on the health benefits of a plant-based diet. So, we suppose that the main reason for the rise of the trend in health care, besides the support of animal rights and environmental protection.

Moreover, there is a pattern of having relatively more tweets on the “vegan” subject during the spring months, which plummet in the middle of the summer and start rising in autumn again; see Fig. 6. The pattern can be explained by the fact that in the winter and spring months, people generally have a vitamin deficiency and start focusing on their health more.

Figure 4 A number of tweets by the query.

Data description

The collected data set consists of 113,677 tweets. In the data set, 2,682 tweets correspond to health topics, containing healthy, wholesome, and full-blooded keywords. Also, there are three levels of vegan topic detailing, namely “vegan-1”, “vegan-4”, and “vegan-7”. The level of “vegan-1” corresponds to just one keyword, namely “vegan”, and there are 32653 tweets of it. The level of “vegan-4” is about “vegan, vegetarian, veggie, veganism,” and there are 53,201 tweets. The level of “vegan-7” is with the following keywords: “vegan, vegetarian, veggie, veganism, plant-based, plant-based, plant based” and there are 65,213 tweets, and 2,075 tweets include keywords about both veganism and health. Also, there are corresponding intersections between the health topic and the levels of the vegan topics: 1,359, 1,791, 2,252 tweets. The methodology of data collecting assumes that tweets are distributed equally in terms of years, months, and weekdays; however, the distribution by hour expects that users wrote the most tweets in the evening.2

A visual representation of the keyword distribution can be seen in Fig. 7 from which it becomes clear which topics and discussions prevail in the presented data set. For example, the following topics: diets, recipes, animal welfare, etc.

We presented the cross-correlations between keywords in Fig. 8. You can see an anti-correlation between the keyword “plant milk” and the other keywords from the image. From the additional analysis, the cause of such phenomena is the insufficiency of plant-milk tweets against other subtopics, which leads to negative correlation values. Also, we should mention that there has been an increasing trend of mentions on Twitter “plant-milk” from 2010 to the present. Also, there is another systematic distortion in the cross-correlation diagram, particularly the high correlation value between vegan and veganism because one of them is a sub-string of another. All other features of the dataset are pretty much inter-correlated.

Figure 5 Tweets having at least one like, by year from the 2010 to 2022 period.

It is interesting to show the analysis of the relationship between health and vegan topics in a time-based manner. In Fig. 9, there is Spearman’s rank correlation between the number of tweets dedicated to these topics.

Figure 6 Tweets having at least one like, by month of a year for the 2010 to 2022 period.

There is a peak correlation with all topics in 2012. We should notice that the correlation with the topic of cruelty-free arises to its maximum with a one-year lag. The lowest maximum owns the keyword “vegan.” The highest maximum owns the keyword “tofu.” The next explicit historical maximum was in 2015; the highest value belongs to the keyword “vegetarian,” and the lowest is the same as in 2012. In 2018 there is another peak with a different leader and outsider, namely the highest value owned to “tofu” and the lowest one to a couple of keywords “plant-based, veggie.” And the lowest peak all the time was in 2020; the leaders of the year are the keywords “plant-based, veganism”, and the outsider is the keyword “tofu”.

Another presented metric of discourse analysis of these topics is mutual information. From Fig. 9 we can see that the peak of simultaneous mentions of keywords from both topics is in 2018. Also, Fig. 9 shows an ascending trend from 2010 to 2018.

We strongly believe that such findings can give a clue to understanding the development of the social discourse on a couple of interrelated topics, “vegan” and “health.”

Pre-processing

Twitter is an excellent source for public opinion mining. Yet, we must deal with its informal language, regular misspelling, wrong word order, etc. Therefore, Twitter data cannot be analyzed in its initial raw form. Twitter users type shortened words to maintain the word length limitation (e.g., 140 characters maximum) in a tweet and tend to use digitally mediated language, so-called Slanguage (Turney, 2002), that incorporates slang, abbreviations that are natural in relaxed and unofficial context. In other words, tweets are required to be cleaned before the analysis processes. This process involves five steps: Cleaning, Case Folding, Tokenization, Slang Lookup, and Stopword Removal.

For better understanding, suppose we have a sample tweet in our data—“I’m glad he responded to you. Too bad he thinks veganism is some kind of perfection or is difficult. Seriously. Difficult? C’mon:)”. The pre-processing steps are explained below and demonstrated in Fig. 10.

Figure 7 Word cloud of the presented dataset with tweets in vegan context.

Cleaning

There are a few specific types of unwanted textual parts that a standard NLTK tokenizer misses: mentions, emoticons, and URLs. At this step, we remove unnecessary text associated with them. The removed parts were: emoticons and punctuation marks.

Case folding

The primary purpose of this step is to convert the tweet to lowercase, thus minimizing the vocabulary size.

Tokenization

The tokenization step was done using NLTK tokenizer (Bird & Klein, 2009). We split all the words separated by a white space and remove all the punctuations and numbers. The tokenization process is based on regular expressions (regexp). Resultant tokens can be seen in Fig. 10.

Slang lookup

At this stage, we have the primary goal to remove slang words from the dataset. The term slang is a novel word that is not standard, and it is a dynamic, innovative, and ever-changing phenomenon. During text analysis, slang words bring unsuitable implications. We apply the synonym method at this step to replace slang with its standard equivalent word (Utama, 2019). We do this by comparing the words in the dataset to the slang word list and synonyms and finding matches. In our example tweet, the slang words C’mon, I’m were modified.

Stopword removal

During the pre-processing stage, stop-word elimination is a crucial step. Not all of the words obtained at the previous stage are significant. Certain words have no sense, and they refer to stop words (articles, conjunctions, some adverbs, etc.). These terms should be removed from the dataset to speed up the processing time (Utama, 2019). Returning to our example, we removed common stop words, like I, am, he, to, you, too, is, of, or, on.

We examined the general structure of a tweet in the preceding explanation and how to pre-process its content. As a result, we have preprocessed tweets in the form of clean tokens that can be further passed to perform mutual information feature selection.

Sentiment detection

Computing term probabilities

Computation of P(t) (the probability of observing the term t) and Pt1∧t2(the probability of co-occurring of the terms t1 and t2) is defined in Eqs. (1) and (2) (Turney & Littman, 2003a; Turney & Littman, 2003b; Turney, 2002). (1) Pt=DFtD,

where D is the number of documents (or tweets in our case) in our dataset, and DF is Document Frequency or number of documents where the term occurs (Bonzanini, 2016). (2) Pt1∧t2=DFt1∧t2D

Mutual information feature selection

By deleting uninformative characteristics, feature selection tries to minimize classification calculation time. Mutual information is a machine learning selection approach demonstrating how important a feature is in making an accurate prediction (Bonzanini, 2016).

We use mutual information to determine the strength of a semantic association between two words. We compute the PMI of each pair of terms and then calculate the semantic orientation given two vocabularies for positive and negative terms. If some word is commonly associated with concepts in the positive lexicon, it has a positive semantic orientation. A schematic representation of the method can be seen in Fig. 11.

Figure 8 Cross-correlation between the features of the data-set.

To begin, we define a word’s semantic orientation (SO) as its positive and negative connections difference (Turney & Littman, 2003a; Turney & Littman, 2003b; Turney, 2002). In practice, we’re looking for a way to determine “how near” a word is to phrases like good and awful. Point-wise mutual information (PMI) is the preferred measure of “closeness,” which is defined in Eq. (3) (Church & Hanks, 2002; Turney & Littman, 2003a; Turney & Littman, 2003b; Turney, 2002). (3) PMIt1,t2=logPt1∧t2Pt1×Pt2,

where t1 and t2 are terms and P designates probability function.

The semantic orientation (SO) of a term t is defined in Eq. (4). (4) SOt= ∑t′∈V+PMIt,t′−∑t′∈V−PMIt,t′,

where V+ and V− are vocabularies of positive and negative words respectively.

Semantic orientations of some context-specific words from our collected dataset are presented in Fig. 12. As we can see from the figure, many words specifying animal products have negative semantic orientations in the vegan context. In contrast, the words associated with plant-based diets mostly have a positive semantic orientation. It is of interest to note that the word “eggs” has a far more negative score than the word “beef”, for example (−1.716 and −0.226, respectively). Note that these terms may have different scores when discussed in non-vegan contexts. At this step, we can update the general lists of positive and negative words we initially had with context-specific words from our dataset that got very high or very low scores, respectively.

Figure 9 The time-based evaluation of mutual information on two topics, health and vegan (left). The time-based correlation between the topic of health and vegan topic keywords (right).

The corresponding Algorithm 1 along with the pseudocode can be seen below. Python code and all supplemented data are also publicly available.3

_______________________  Algorithm 1: Mutual information for feature selection                        ____     Data: tweets in csv file     Result: obtained PMI for semantic orientation calculation;     df_clean[”tokens”] = tokenized tweets by words;     remove stopwords and change slang words;     Function build_frequencies(tweets):         word_count = calculated count of each word;    word_pairs = calculated word pairs count;    return (word_count, word_pairs);     Function build_probabilities(word_count, word_pairs,     tweets_count):         word_probability = dictionary that contains calculated probability of    each word;    word_pair_probability = dictionary that contains calculated    probability of each pair;    return (word_probability, word_pair_probability);     Function build_pmi(tweets):         word_count,word_pairs = build_frequencies(tweets)    word_probability,word_pair_probability =    build_probabilities(word_count,word_pairs,tweets_count)    pmi ← emptydictionary for term1 in word_probability do        for term2 in word_pairs[term1] do    denom = word_probability[term1] * word_probability[term2]    pmi[term1][term2] calculated by formula 1    end    end    return pmi, word_count;     // Load set of positive and negative words    Function get_semantic_orientation(word):         pmi, word_count = build_pmi(df_clean[”tokens”])    if word in word_count then        return value calculated by Formula 2    end

Classifying tweets

Furthermore, we can find a tweet sentiment by finding a semantic orientation of cleaned tokens from each tweet in a training dataset. The sentiment score of a tweet is calculated as the sum of the sentiment scores of each term present in the tweet and in the lexicon (Eq. (5)) (Bindal & Chatterjee, 2016): (5) TweetScoreT= ∑t∈TSOt,

where T is a tweet, t - specific term in a tweet, SO(t) is a term score (semantic orientation of t, Eq. (4)). Note that we do not consider the scores for out-of-lexicon terms. The classification threshold for neutral tweets was set from −0.1 to 0.1. The tweets with negative scores were assigned a negative sentiment, whereas positively scored tweets got a positive sentiment. The scores close to 0 represent neutral tweets. It happens when the PMIs for positive and negative balance out.

The accuracy of the proposed method was 0.62 on a training dataset (labeled tweets from Kaggle). This indicates the goodness of the sentiment lexicon, considering that neither this dataset was manually annotated by humans nor the classifier used for labeling was trained on vegan-related language. We will continue discussing the accuracy in the Results section when we utilize a dataset with vegan-related tweets labeled by humans.

Emotion detection

Emotion recognition is frequently considered a subtask of polarity detection (Cambria, 2016), and it became the next step naturally after the basic sentiment task (Ashraf et al., 2022). A user puts into the tweet not just a positive or negative meaning but often a range of emotions. Other users can easily recognize and experience these emotions in texts. Emotion investigation can help us better understand opinions rooted in tweets. Detection of emotions behind the tweets was done using text2emotion 0.0.5 (Python Software Foundation, 2020). Five different emotion categories were used, such as Happy, Angry, Sad, Surprise and Fear. Text2emotions returns the dictionary with scores for each emotion category.

We could assume that the tweet belongs to a particular emotion if the score for that emotion is high. However, we don’t want to miss emotions of average and weak power. So we use a fuzzy approach (Zadeh, 1965) to manipulate emotions expressiveness.

A class of objects with a continuum of grades of membership is referred to as a fuzzy set (Zadeh, 1965; Zadeh, 1975). Several studies used a fuzzy approach to represent the strength or type of emotions and differentiate between different emotional labels (Matiko, Beeby & Tudor, 2014; Karyotis et al., 2016; Qamar & Ahmad, 2015).

The primary motivation for using fuzzification in our study is that it allows us to judge emotions in a human-consistent style. Since emotions don’t have clearly defined boundaries, fuzzy sets are very useful for dealing with these concepts. By partitioning the range of feasible emotions corresponding to linguistic tags, we get more intuitive and human-consistent output (Shamoi et al., 2016). Likewise, fuzzy sets can easily depict the gradual change from one emotion label to another can (Zadeh, 1975). A language value is more similar to human cognitive processes than a number, despite being less precise (Shamoi & Inoue, 2012).

We represent Emotion power as an ordered list of terms of the fuzzy (linguistic) variable X = “Emotion power” by means of primary linguistic terms L = {Weak, Medium, Strong}, specifying its level. Figure 13 below depicts the fuzzy sets for the Emotion power. For the sake of simplicity, we use triangular membership functions (Eq. (6)) representing the linguistic labels expressing the fuzzy variable Emotion power for each of the emotion categories. It is good enough to catch the ambiguity of the linguistic assessments. Such fuzzy partition was made based on subjective perception —we just ran a few experiments and adjusted the fuzzy sets. In the future, we can improve the partition by conducting a survey based on human emotion categorization (Shamoi, Inoue & Kawanaka, 2014).

Figure 10 Sample tweet.

The pre-processing steps.

(6) μax=0,ifx≤a.x−ab−a,ifa≤x≤b.c−xc−b,ifb≤x≤c.0,ifx≥c,

where a, b, c represent the x coordinates of the three vertices of μa(x) in a fuzzy set A. Although we have five types of emotions, not all of them are present in a tweet. We take into account only those that obtained a non-zero score. For example, some specific tweet may contain weak sadness and strong surprise emotions only.

In the case of composite emotion impressions, formed from atomic ones with the help of various connectives (and, or) (Shamoi & Inoue, 2012) and hedges (e.g., not, very, more-or-less), general formulas from fuzzy theory, like α-cut, reinforcing and weakening modifiers, set operations (Shamoi, Inoue & Kawanaka, 2020) are used. α-cut is a crisp set that includes all the members of the given fuzzy subset whose values are not less than α for 0 < α ≤ 1. The α-cut of A is defined as Zadeh (1965): (7) Aα=x∈X|Ax⩾α.

These basic formulas allow us to retrieve some interesting tweets and judge emotions in human-consistent style, like “extremely happy and surprised”, “very very angry but not sad”, etc.

As we see, emotion analysis can help to detect more complex feelings embedded in a tweet. For example, ‘surprise’ is an emotion that can express both positive and negative sentiments (Sailunaz & Alhajj, 2019). The negative form of surprise is when you eat well, yet the tests reveal nutrient deficiency. On the other hand, the positive emotion of surprise is when you are surprised by an extensive choice of vegan options in a restaurant.

Let us take, for example, a tweet “Sorry Vegans, steak taste good.”. Its sentiment score is Neutral (−0.043) due to its balance of positive and negative words. At the same time, emotion analysis results are ‘Happy’: ‘Medium’, ‘Sad’: ‘Weak’, ‘Fear’: ‘Weak’. So, detecting emotions can be particularly useful for balanced tweets that standard SA can neutralize.

Python code and all supplemented data for emotion detection is publicly available.4

Dataset annotation on an 11-point scale

We initiated the annotation to evaluate the method because there are no available vegan-specific datasets labeled by humans or by a classifier trained in the vegan context. The trial dataset consisted of 100 tweets and was created using random sampling from our vegan-related unlabeled dataset described before.

Five humans performed the annotation process. There were three vegans and two omnivores among them. Data annotation should be performed by someone with a thorough understanding of the topic, so local vegans with 15, 5, and 2 years of vegan experience were asked to contribute. Vegans were given double weight since they are more familiar with the lexicon of the context. The human raters used separate Google sheets they got online to label the sentiment polarity of the tweets. A screenshot of the form (Google Sheets) is provided in Fig. 14.

Figure 11 Schematic representation of the model.

Each tweet in a dataset was assigned a category by all five annotators. A test dataset was annotated using an 11-point scale annotation method described in Ghosh et al. (2015). Each annotator was instructed to assign a score to an opinion in a tweet based on a perceived meaning between −5 (representing strong displeasure) and +5 (for strongly positive tweets). Details can be seen in Fig. 14.

We used tactics given in Ghosh et al. (2015)—If 60% or more of annotator labels are assessed to be outliers, then the annotator judgments are dismissed from the task. We use this formula Eq. (8) to identify if a judgement Ai,j is an outlier: (Ghosh et al., 2015): (8) |Ai,j−avgAi′,j|>stdttj,

where stdt(tj) is the standard deviation of all scores given for a tweet tj.

After the weighted average calculation, we revealed one annotator whose proportion of outliers was 71%, so his annotations were excluded from the analysis, and the weighted average for each tweet was computed again.

Eventually, based on weighted average scoring, each tweet in a trial dataset obtained either a positive, neutral, or negative score (see Table 3). The annotation procedure resulted in the following sentiment distribution: positive sentiments (57%), neutral sentiments (0%), and negative sentiments (43%), with classification thresholds set at −0.1 and +0.1.

Experimental Results and Discussion

Sentiment detection results

Some tweets lead to discussions involving a lot of users. Popular tweets are tweets with relatively high responses in the form of retweets, comments, and likes (Zubiaga et al., 2014). For sentiment analysis, we selected popular tweets from our vegan dataset for each year from 2010 to 2022. By popular tweets, we mean those tweets that were retweeted at least once, indicating that the opinion expressed in them was supported by other users.

Obtained sentiment analysis results are presented in Fig. 15 and Table 4. Results indicate that positive sentiment in popular tweets consistently dominates over the years. It can also be seen that all sentiments started to grow, particularly during those years when the associated studies on the health benefits of a plant-based diet appeared. As for 2022, we see a fall in the number of popular tweets, which is associated with the fact that our dataset included only 4.5 months for 2022. However, it is evident that in 2022 the proportion of tweets with negative sentiment relatively decreased compared to previous years and is nearly equal to the proportion of neutral tweets. This could be interpreted in several ways, the most common of which is the impact of COVID-19 and global warming. Plant-based foods are becoming more common in post-pandemic diets. The investigation of the ‘Popular Tweets’ column of Table 4 reveals that levels of interest in plant-based diets have increased from 2010 to 2022. Note that both Fig. 15 and Table 4 focus only on tweets that have been retweeted at least once (RT ≥ 1).

Figure 12 Semantic orientations of some context-specific words.

This illustration can give us an idea of the main topics/ words when veganism is discussed in a positive or negative context. As can be seen, there is plenty of healthy food words which are scored as positive in vegan tweets.

Table 5 shows examples of some positive, neutral, and negative tweets from our dataset along with their computed scores using Eq. (5).

Emotion detection results

Sentiment analysis helped us to identify that positive sentiments were prevailing. However, what are the leading emotions behind this growing interest, and how were they changing throughout the period?

The graphs in Fig. 16 show the comparisons of emotions across years that indicate some interesting facts. The release of a lot of popular documentary movies revealing violence towards animals in agriculture and the health benefits of plant-based diets can explain the spike in all emotions in 2018 (What the Health, 2017 (7.2/10 IMDb); Game Changers, 2018 (7.8/10 IMDb); Dominion, 2018 (9/10 IMDb); Eating You Alive, 2018 (7.9/10 IMDb); 73 cows, 2018 (7.8/10 IMDb); Let Us Be Heroes, 2018 (8.7/10 IMDb)). Moreover, in 2018, UN IPCC reported that climate change catastrophe would occur in the upcoming 12 years (Masson-Delmotte et al., 2019).

Figure 13 Fuzzification of emotions levels.

The growth of Happy emotion, which we can clearly observe from 2018, is possibly connected with the fact that more and more vegan options, products, and places are starting to open, and people feel joyful about their health, lifestyle, and food. The ‘surprise’ emotion noticeably increased from 2018 as well, which can be connected to the fact that documentaries about the benefits of the vegan diet and the truth about animal abuse premiered this year (What the Health, 2017 (7.2/10 IMDb); Game Changers, 2018 (7.8/10 IMDb); Dominion, 2018 (9/10 IMDb); Eating You Alive, 2018 (7.9/10 IMDb); 73 cows, 2018 (7.8/10 IMDb); Let Us Be Heroes, 2018(8.7/10 IMDb)).5 This fact can also be considered as the reason for the rise of the Sad emotion because most people feel heartbroken, frustrated, and sad about realizing the heavy burden that animals inhabit for just one human’s meal. Furthermore, on the graph, we can recognize that emotions of Fear were mostly medium and strong throughout the period, which contrasts sharply with other types of emotions. This can be explained by the fact that many people tend to believe that a vegetarian or vegan diet can lead to nutritional deficiency (Paslakis et al., 2020).

Figure 17 demonstrates how individuals perceived veganism over time. We investigated that the public’s interest in plant-based diets has been growing noticeably since 2010. In 2017-2022, compared to 2011-2016, it can be noticed that the number of weak happy and weak angry tweets doubled. At the same time, there were no significant changes in the amounts of medium happy, angry and strong happy tweets. But it should be noted that there were fewer strong angry tweets by 2017–2022. From this, it can be assumed that by 2017–2022, people’s attitude to plant-based lifestyles was changing for the better.

Figure 14 Sample form (Google Sheets) with tweets for annotation that was provided separately to each annotator.

Score descriptions and data annotation method were taken from Ghosh et al. (2015). The form contains 100 random tweets from our unlabeled vegan dataset.

Table 6 illustrates examples of emotions identified in tweets from our dataset.

We also performed a basic analysis based on the usage frequency of words in a vegan context across 12 years. The bar chart (see Fig. 18) illustrates the most frequently used terms (excluding keywords we used for our dataset). It can be clearly seen that when people discuss veganism, they usually don’t just talk about food (meat, products, recipe, organic, delicious), but also touch on the topics of animals, health, and beauty (perhaps this is related to the topic of testing products on animals).

Table 3 Examples of sentiment results for some tweets based on manual annotation.

Evaluations provided by Annotator A3 were excluded from the weighted average calculation as the number of outliers in its results exceeded 60%. Annotators with asterisks were given a double weight because they are vegans and are good at a context-specific lexicon.

#	Tweet	A1* w = 2	A2 w = 1	A3 (Excluded)	A4* w = 2	A5* w = 2	Weighted AVG	Annotation result	Sentiment	
1	Moral consis...	1	3	5	1	4	2.14	1	Positive	
2	I don’t care ...	−3	1	3	−2	−4	−2.43	−1	Negative	
3	Coke Bottle ...	3	4	5	2	3	2.86	1	Positive	
4	A great revie...	3	4	0	2	3	2.86	1	Positive	
5	Over 110M a...	−3	0	5	−2	−3	−2.29	−1	Negative	
6	What’s more...	−5	0	5	−2	−2	−2.57	−1	Negative	

Figure 15 Sentiment analysis results for popular tweets (retweeted at least once, RT ≥ 1) from 2010 to 2022.

Table 4 Sentiment analysis results for popular tweets (retweeted at least once, RT ≥ 1) from 2010 to 2022.

Year	Popular tweets	Positive	Positive, %	Negative	Negative, %	Neutral	Neutral, %	
2010	414	200	48.31	142	34.3	72	17.39	
2011	542	262	48.34	163	30.07	117	21.59	
2012	745	322	43.22	236	31.68	187	25.1	
2013	927	443	47.79	291	31.39	193	20.82	
2014	1,265	501	39.6	384	30.36	380	30.04	
2015	1,254	527	42.03	364	29.03	363	28.95	
2016	1,409	654	46.42	380	26.97	375	26.61	
2017	1,397	651	46.6	412	29.49	334	23.91	
2018	1,257	588	46.78	366	29.12	303	24.11	
2019	1,407	643	45.7	449	31.91	315	22.39	
2020	1,253	571	45.57	391	31.21	291	23.22	
2021	1,227	615	50.12	389	31.7	223	18.17	
2022	448	203	45.31	132	29.46	113	25.22	

Table 5 Examples of some positive, neutral, and negative tweets from our dataset along with their computed scores.

Tweet	Score	Sentiment	
“Sitting in a vegan cafe in Montreal called aux vivres! Everyone is under 25 Which makes me feel old but encouraged for future of veganism!”	1.65	Positive	
“You can’t convince anyone of the benefits of vegetarian/veganism when your diet consists of multiple fake meats. Just admit you want meat.”	−0.15	Negative	
“Stupid Questions (and clever answers) about veganism never gets old”	−0.27	Negative	
“My boss wants to make me a vegan soy burger. Whoa slow down there.”	−0.02	Neutral	
“Enjoying the vegan treats at Café Petit Gteau”	0.72	Positive	
“#Flexitarian diet: mostly plant-based with small portions of meat, fish and poultry.”	0.02	Neutral	
“my healthy goals are to lose 10 pounds, drink more water and use more plant-based foods.”	0.9	Positive	

Our findings support the results of the effects of sentiment on information diffusion, obtained by Ferrara & Yang (2015). According to their findings, positive posts reach a wider audience, implying that people are more likely to retweet positive content, a phenomenon known as positive bias. Discussions with more than 200 tweets are considered active ones (Ferrara & Yang, 2015). We filtered out only active, extremely popular tweets having at least 200 retweets (RT ≥ 200). Next, we fit the data within a polynomial function as depicted in Fig. 19. It illustrates the number of retweets collected by the tweets as a function of the sentiment score expressed therein. As seen from the figure, among such viral tweets, we have primarily neutral and slightly positive tweets, but not extremely positive ones. Nevertheless, the average number of retweets is higher for negative tweets (0.706, 0.503, and 0.369 for negative, positive, and neutral tweets, respectively).

Figure 16 Prevailing emotions in tweets across years.

Figure 17 Bar charts for happy and angry emotions for a 5-year subsequent period.

Table 6 Examples of fuzzy emotion detection in tweets from our dataset.

Sample tweet	Crisp emotions	Fuzzified emotions	
“Always puzzles me why so many plant based menu items reference non-plant based foods.”	{‘Happy’: 0.0, ‘Angry’: 0.0, ‘Surprise’: 0.2, ‘Sad’: 0.6, ‘Fear’: 0.2}	{ ‘Surprise’: ‘Weak’, ‘Sad’: ‘Medium’, ‘Fear’: ‘Weak’}	
“handsome, musician, tattooed and vegetarian I’m already in love”	{‘Happy’: 1.0, ‘Angry’: 0.0, ‘Surprise’: 0.0, ‘Sad’: 0.0, ‘Fear’: 0.0}	{‘Happy’: ‘Strong’}	
“Meat is bad for you. Emissions from agriculture are projected to increase 80% by 2050. Being vegan makes a difference.”	{‘Happy’: 0.0, ‘Angry’: 0.0, ‘Surprise’: 0.0, ‘Sad’: 0.67, ‘Fear’: 0.33}	{‘Sad’: ’Medium’, ‘Fear’: ‘Medium’}	
“So if you don’t want to shop cruelty free for the animals, do it for your own well being”	{‘Happy’: 0.2, ‘Angry’: 0.4, ‘Surprise’: 0.0, ‘Sad’: 0.4, ‘Fear’: 0.0}	{‘Happy’: ’Weak’, ‘Angry’: ‘Medium’, ‘Sad’: ‘Medium’}	
“I became a vegetarian in college when I got very sick from eating meat. I really don’t miss eating meat.”	{‘Happy’: 0.0, ‘Angry’: 0.0, ‘Surprise’: 0.17, ‘Sad’: 0.33, ‘Fear’: 0.5}	{‘Surprise’: ‘Weak’, ‘Sad’: ‘Medium’, ‘Fear’: ‘Medium’}	
“I Thought u were a vegetarian”	{‘Happy’: 0.0, ‘Angry’: 0.0, ‘Surprise’: 1.0, ‘Sad’: 0.0, ‘Fear’: 0.0}	{‘Surprise’: ‘Strong’,}	

Figure 18 Top frequent words in a dataset (excluding search keywords).

Figure 19 Extremely popular tweets (RT ≥ 200) and their sentiment values.

Accuracy evaluation

We use popular sentiment analysis models VADER (Honnibal & Montani, 2017) and spaCy (Honnibal & Montani, 2017) and the human-annotated dataset discussed earlier to evaluate the performance of our approach.

Table 7 presents the Mutual information scoring function against VADER and spaCy scorings along with the sample tweets from the labeled dataset.

Table 8 shows the three-class classification accuracy for each method. From Table 8 we can see that our classifier (0.77) performs better than VADER (0.69) and spaCy (0.67) for the selected context, but still not as well as individual human rater with special context knowledge (0.87). Unsurprisingly, our scoring method obtained much higher accuracy (0.77) than the accuracy on the training data set labeled automatically without accounting for the vegan-tweets-based lexicon. Although we know that VADER is sensitive to social media lexicon, its accuracy decreases since we use it in domain-specific vegan tweets. The dataset we used for training has narrow variability and context-specific vocabulary, so a PMI-based classifier performs better.

Table 7 Examples of sentiment classification results of VADER, spaCy, and PMI-based sentiment analysis for the manually annotated dataset containing 100 tweets.

Tweet	VADER Sent.	spaCy Sent.	PMI-based Sent.	PMI-based Score	
Moral consistency requires veganism. Go vegan today!	Neutral	Neutral	Positive	0.164	
I don’t care what our desperate primate ancestors ate to *survive.* Real men do not murder weak, helpless, little creatures.	Positive	Negative	Neutral	0.009	
Coke Bottle Redesign: Stackable, space friendly and 100% plant-based	Positive	Positive	Positive	1.096	
A great review from MyBeautyBunny! Dr. Sharp’s Cruelty-Free Dental Care Solution	Positive	Positive	Positive	1.462	
Over 110M animals r killed in the food industry every single day. Can u really put off going vegan or talking 2 others about veganism?	Negative	Negative	Negative	−0.493	
What’s more like God? Wants you to enslave, breed and murder innocent beings? Or protect them from those acts?	Negative	Positive	Negative	−0.514	

Table 8 Sentiment prediction accuracy for VADER, spaCy, PMI-based classifiers. We also provided accuracy for annotator A4, just for comparison.

	VADER Sent.	spaCy Sent.	PMI-based Sent.	A4 Sent.	
Accuracy	0.69	0.67	0.77	0.87	

Next, let us look at how well different classifiers agree on the sentiments of the tweets. We calculated the intersections of each method’s positive or negative proportions to compute the agreement. The Agreement Score represents the fraction of tweets both methods agree (have the same predicted polarity) (Gonalves et al., 2013). Table 9 shows the percentage of agreement between methods (phenomenon of methods agreement is discussed in Gonalves et al. (2013)).

Table 9 Percentage of agreement between our method, VADER, and spaCy.

	VADER Sent.	spaCy Sent.	PMI-based Sent.	
VADER Sent.	–	69	63	
spaCy Sent.	69	–	55	
PMI-based Sent.	63	55	-	

From Table 9 we can see that our classifier has a much higher agreement with VADER than with spaCy (63% and 55% respectively). Pearson’s rank correlation between our method and VADER came out to be 0.565, which indicates that they are sensibly correlated.

Discussion

Let us consider how the current research results compare to other studies’ results. Previously, there was similar research, but mostly on much broader topics, such as health, sports, and a healthy diet. We consider a narrower topic and study not only sentiments but also emotions for such a lengthy 12-year period. Such a study using Twitter data can confirm and supplement existing survey data on the subject and, in certain cases, even replace it. Our findings confirm previous studies that have reported a growth in the vegan trend (Craig, 2009; López et al., 2019; Pila, Kvasnikov Stanislavsk & Kvasnika, 2021; Cooper, 2018). For example, according to the study by Cooper (2018), climate catastrophe makes more people become vegan and reject animal speciesism.

Previous studies have shown that the number of people adopting a vegan or vegetarian diet is increasing (López et al., 2019). Another study revealed that Twitter users consider vegan, organic, and homemade foods the healthiest options (Pila, Kvasnikov Stanislavsk & Kvasnika, 2021). The other study by Faunalytics (2021) used Twitter to analyze how interest in the vegan diet has varied from 2019 to 2020. Our findings agree that vegan-related tweets are more often used in a positive context than in a neutral or negative context. Next, the results of the study by Lawo et al. (2020) show that most people became vegan because they saw the reality of animal suffering on YouTube, Netflix documentary videos, and shows. Our findings confirm this, since we witnessed a clear spike in all emotions in the years of the documentaries release, especially ‘Surprise’.

According to research, a vegan diet is especially popular among teenagers and youth (Paslakis et al., 2020; Craig, 2009), so, probably, the drivers of the trend are primarily young people, who spend a lot of time on social networks (Castillo de Mesa et al., 2020).

We identified that the ‘Fear’ emotions are also present. Moreover, they are growing, probably due to the risk of certain nutritional deficiencies that may be present if the vegan diet of an individual is not balanced (Craig, 2009) (not a whole food plant-based diet). Our findings support the earlier result obtained by Twine (2017). They conducted a survey in the UK and identified that while transitioning to a vegan diet, people have fear because of a lack of nutritional knowledge. The same result was obtained by Jennings et al. (2019). A vegan diet can indeed lead to deficiencies if it consists of French fries, bread, and vegan candy. These are vegan but unhealthy foods. A whole, plant-based diet rich in vegetables, fruits, whole grains, beans, legumes, seeds, and nuts needs to be promoted (Willett et al., 2019).

In addition, our results corroborate the findings of the study on the influence of sentiment on information diffusion (positivity bias) (Ferrara & Yang, 2015).

Conclusion

The vegan diet has been shown to provide a variety of health and environmental benefits in multiple studies. This study aimed to identify how the attitude toward a plant-based diet has been changing from 2010 to 2022 using Twitter social network data. We examined the prevailing sentiments and emotions embedded in tweets and found that veganism is currently frequently debated on social media compared to previous years. As the results suggest, there is a growing acceptance of veganism, the popularity of the vegan diet is expanding, and associated sentiments and emotions are becoming more positive. However, feelings of fear are also present.

Our findings place new information in the public domain, which has significant practical implications. Obtained results can be used to develop corresponding government action programs to promote healthy veganism and reduce the associated feeling of fear among the population. By gaining a deeper understanding of the public perception of veganism and using the substantial research unveiling the health benefits of the vegan diet (Radnitz, Beezhold & DiMatteo, 2015; Willett et al., 2019; Norman & Klaus, 2019; Qian et al., 2019; Tran et al., 2020; Turner-McGrievy, Mandes & Crimarco, 2017; Park & Kim, 2022), clinical practitioners and public health experts can develop efficient health programs and strategies and encourage more people to either follow a healthy vegan diet or reduce animal product consumption. In addition to health benefits, this can help to reduce the impact of climate change (Dagevos & Voordouw, 2013; Willett et al., 2019; Scarborough et al., 2014; Masson-Delmotte et al., 2019; Cleveland & Gee, 2017; Chai et al., 2019; Eisen & Brown, 2022). Even slight decreases in dairy and meat consumption by many people could make a significant contribution to climate change (Scarborough et al., 2014). Results also give us a sense of idea where this trend is heading because having a positive attitude is the first step toward moving on a plant-based diet. The main conclusions of our study are presented in Table 10.

Table 10 The main findings of the study.

Conclusion	Evidence and details	
Sentiments and emotions in vegan-related tweets are becoming more positive.	See Fig. 15, Table 4, Figs. 16, 17	
Positive sentiments in tweets are consistently dominating.	See Fig. 15, Table 4	
Fear emotions were mostly medium and strong throughout the period.	See Fig. 16	
There was a clear spike in all emotions in the years of the documentaries release, especially Surprise.	See Fig. 16	
Sentiments started to grow during those years when the associated studies on the benefits of a plant-based diet appeared	See Fig. 15, Table 4	
In 2022 the proportion of negative tweets relatively decreased compared to previous years. Possibly, due to the impact of COVID-19 and global warming, plant-based foods are becoming more common in post-pandemic diets.	See Fig. 15	
Positive posts reach a wider audience (positive bias)	See Fig. 19	
The PMI-based scoring method obtained higher accuracy than VADER and spaCY for vegan-specific tweets.	See Table 8	

The method we proposed was cross-validated on an annotated subsample of a dataset and achieved good accuracy (0.77). Our scoring method obtained much higher accuracy than VADER and spaCY for vegan-specific tweets. The main reason is that the dataset we used for training had narrow variability and context-specific vocabulary.

The study has certain limitations. Our analysis keywords are limited, but we plan to extend them in our future works. Next, the estimates of public perception of veganism were based on only one social network (Twitter); in future works, we plan to include other social media, namely Instagram and Facebook. We also plan to explore vegan tweets corresponding to the earlier period as well. Moreover, in this work, we focused on English tweets. However, the attitude toward a plant-based diet differs from country to country. Although English is popular in many countries, analyzing mainly English tweets may not accurately reflect the overall sentiment worldwide. For example, in Post-Soviet countries, acceptance and rise of veganism go much slower due to the absence of vegan communities and limited access to vegan products (Aavik, 2019). The popularity of avoiding or limiting meat products varies greatly by country, so it is important to consider different regions. Fellow researchers might now extend this study by exploring the public perception of veganism across different regions.

Supplemental Information

Supplemental Information 1 Annotated dataset

Click here for additional data file.

Additional Information and Declarations

Competing Interests

Author Contributions

Data Availability

1 The data collection Python script is available at https://github.com/iskander-akhmetov/vegan_tweets_SA.

2 The data description Python script is available at https://github.com/AlexPak/VeganTwitterEDA/blob/main/Visualization.ipynb.

3 Sentiment analysis Python code and all supplemented data is available at https://github.com/elvinast/Sentiment-Analysis-Vegan-Tweets

4 Python code and all supplemented data is available at https://github.com/elvinast/Sentiment-Analysis-Vegan-Tweets.

5 Information is taken from official IMDB site https://www.imdb.com/.

The authors declare there are no competing interests.

Elvina Shamoi conceived and designed the experiments, performed the experiments, analyzed the data, performed the computation work, prepared figures and/or tables, and approved the final draft.

Akniyet Turdybay conceived and designed the experiments, performed the experiments, analyzed the data, performed the computation work, prepared figures and/or tables, and approved the final draft.

Pakizar Shamoi conceived and designed the experiments, performed the experiments, analyzed the data, performed the computation work, prepared figures and/or tables, authored or reviewed drafts of the article, and approved the final draft.

Iskander Akhmetov performed the experiments, analyzed the data, performed the computation work, prepared figures and/or tables, authored or reviewed drafts of the article, data scraping to prepare dataset, and approved the final draft.

Assel Jaxylykova analyzed the data, performed the computation work, prepared figures and/or tables, and approved the final draft.

Alexandr Pak analyzed the data, performed the computation work, prepared figures and/or tables, and approved the final draft.

The following information was supplied regarding data availability:

The annotated data is available in the Supplemental Files.

The data is available at GitHub: https://github.com/iskander-akhmetov/vegan_tweets_SA.

The data visualization is also available at GitHub: https://github.com/AlexPak/VeganTwitterEDA/blob/main/Visualization.ipynb.

The code is available at GitHub: https://github.com/elvinast/Sentiment-Analysis-Vegan-Tweets.

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
