# Peer review of "Sentiment analysis of vegan related tweets using mutual information for feature selection"

_PeerJ Computer Science, doi:10.7717/peerj-cs.1149_

## Round 0.1 · original submission · Major Revisions

The topic is interesting. The scientific soundness needs improvement following the comments from the reviewers. There are methodological weaknesses and a need for further reinforcement of that section. As well as presentation of the results and discussion sections.

Carefully implement reviewer comments and resubmit your work.

Reviewers have suggested that you cite specific references. You are welcome to add it/them if you believe they are relevant. However, you are not required to include these citations, and if you do not include them, this will not influence my decision.


·

Basic reporting

In the introduction, the authors must put their research topic in context and to do so, it is useful to provide more data that allows the reader to put the research topic in context. In this sense, authors are asked to include information on the evolution of the use of the social network Twitter, or of veganism, using statistical data, graphs, or a report on these aspects that allows us to see why this topic of study is important and current.
Authors are asked to cite the statement: “However, in recent years , the Covid-19 pandemic and global warming have increased the awareness of plant-based lifestyle” y hagan referencia a qué estudio se refieren con esa información. Así como esta: “In these days of climate change, this subject is becoming increasingly significant and is frequently debated on social media”
On the other hand, the authors indicate on several occasions "Several studies aimed at (...) COVID-19", as in line 52, but do not include references to the studies they refer to in relation to COVID-19. The authors are asked to revise this.
Likewise, in line 67, where the authors comment on the use of social networks, it would be useful to include data on this use to justify what the authors are mentioning and to put the reader in context. In line with this comment, the sentence “Twitter is one of the most widely used social media, that allows users to write everything that comes to their mind in the form of tweets” in line 72 the authors must cite it.
In the Theoretical background section, authors should include citations in the information provided from line 102 to 109, 128-130.
And correct the authors' citations on lines 156, 60.

Experimental design

The Theoretical background section is intended to include previous research work related to the authors' research and also to indicate why that work is relevant to this one. Other authors such as Reyes-Menendez, A., Saura, J. R., & Alvarez-Alonso, C. (2018). Understanding# WorldEnvironmentDay user opinions in Twitter: A topic-based sentiment analysis approach. International journal of environmental research and public health, 15(11), 2537 have carried out studies on the social network Twitter and sentiment analysis.
Authors are also asked to include the hypothesis or research question, i.e., what was the problem or gap that has been detected and about which we have made a prediction that later, in the methodology, we have been able to corroborate or not. In addition, authors are asked to bear in mind that the hypotheses must be based on the theoretical framework.
Authors should pose a research question or formulate a hypothesis and answer it in a clear and well-presented manner, identifying the research gap.
In terms of methodology, the authors do not indicate on which previous studies they have based the use of their methodology to carry out the research. They do not justify the choice of methodology for this study with any previous studies.
The authors are asked to improve the aesthetic presentation of the results in Figure 13 and to place all the tables and figures in the section "Experimental results and discussion" in relation to the text in an orderly way.
Include examples of the documentaries referred to in this sentence: The Surprise emotion noticeably increased from 2018 as well, which can be connected to the fact that documentaries about the benefits of the vegan diet and the truth about animal abuse were premiered this year (line 419).
On the other hand, authors are asked to explain the meaning of Fuzzification.
Authors are encouraged to include references to recent studies on the topics covered such as those carried out by: Reyes-Menendez, A., Saura, J. R., & Filipe, F. (2020). Marketing challenges in the# MeToo era: Gaining business insights using an exploratory sentiment analysis. Heliyon, 6(3), e03626. Or Saura, J. R., Herráez, B. R., & Reyes-Menendez, A. (2019). Comparing a traditional approach for financial Brand Communication Analysis with a Big Data Analytics technique. IEEE Access, 7, 37100-37108.
Although the results are well explained, their presentation is not entirely clear to the reader. It is recommended that the authors produce a summary table with the main conclusions obtained.

Validity of the findings

On the other hand, the paper should present a Discussion section where the results obtained should be compared with previous research to see if the research is supported by what other authors have researched. Authors are asked to include this section following this commentary.
Authors tend to make many claims without basing them on studies or research to support their claims.
Limitations of the research, such as the fact that only one social network is analysed, the number of keywords analysed or the time period included in the analysis, should be included.
The research topic presented by the authors is interesting, but requires the suggested improvements.

Reviewer 2 ·

Basic reporting

This paper aims to utilize Twitter posts to better understand public opinión regarding the vegan (plant-based) diet. The purpose of sentiment analysis is to determine whether a piece of text (tweet in our case) conveys a negative or positive viewpoint
In my opinion, this manuscript is a great work, but needs improve. The key points for review are:

1.- Please use different terms in the "Title" and the "Keywords".
2.- Authors should rewrite the abstract following a structure that responds to the questions:
Value-originality, objectives, method, findings and conclusions.
3.- The structure of the paper is confusing. The theoretical background section should be divided into two. On the one hand, approaches for sentiment analysis and on the other, related works. This second subsection should show the reader the importance of the technique and its various applications to other fields.
4.- In this second subsection authors must add works that have studied sentiment analysis. Authors should incorporate important improvements. It is suggested to add articles entitled:
Saura, J.R., Palos-Sanchez, P.R. and Rios Martin, M.A. (2018). Attitudes to environmental factors in the tourism sector expressed in online comments: An exploratory study. International Journal of Environmental Research and Public Health 15(3), 553; doi:10.3390/ijerph15030553
Serrano, L., Ariza-Montes, A., Nader, M., Sianes, A., & Law, R. (2021). Exploring preferences and sustainable attitudes of Airbnb green users in the review comments and ratings: A text mining approach. Journal of Sustainable Tourism, 29(7), 1134-1152.
Ríos-Martín, M. Á., Folgado-Fernández, J. A., Palos-Sánchez, P. R., & Castejón-Jiménez, P. (2020). The Impact of the Environmental Quality of Online Feedback and Satisfaction When Exploring the Critical Factors for Luxury Hotels. Sustainability, 12(1), 299.
Saura, J. R., Reyes-Menendez, A., & Palos-Sanchez, P. (2019). Are Black Friday Deals Worth It? Mining Twitter Users’ Sentiment and Behavior Response. Journal of Open Innovation: Technology, Market, and Complexity, 5(3), 58
Sánchez, P. R. P., Folgado-Fernández, J. A., & Sánchez, M. A. R. (2022). Virtual Reality Technology: Analysis based on text and opinion mining. Mathematical Biosciences and Engineering, 19(8), 7856-7885.

Experimental design

5.- Improve the quality of figure 1.
6.- The authors continually refer to Covid-19. I wonder if they should include it in the title.
7.- It is suggested to compare the results of the present research with some similar studies which is done before.
8.- Consider the length of the conclusions and divide the conclusions into theoretical and practical
9.- Add DOI for all references.

Validity of the findings

It is suggested to compare the results of the present research with some similar studies which is done before.

---

## Round 0.2 · Minor Revisions

There are minor revisions to be conducted such as a proofread, clarify the implications of your research in the context of the existing literature.

Reviewer 2 ·

Basic reporting

OK

Experimental design

OK

Validity of the findings

OK

Additional comments

OK

---

## Round 0.3 · accepted · Accept

After reviewing the work implemented to increase the scientific soundness of this paper, I consider the research ready to be published.